# High Prevalence of SARS-CoV-2 in an Indigenous Community of the Colombian Amazon Region

**DOI:** 10.3390/tropicalmed6040191

**Published:** 2021-10-27

**Authors:** Héctor Serrano-Coll, Hollman Miller, Camila Rodríguez-Van Der Hamen, Bertha Gastelbondo, Wilkhen Novoa, Misael Oviedo, Ricardo Rivero, Evelin Garay, Salim Mattar

**Affiliations:** 1Instituto de Investigaciones Biológicas del Trópico, Universidad de Córdoba, Montería 230001, Colombia; hectorserranocoll@gmail.com (H.S.-C.); berthagastelbondop@correo.unicordoba.edu.co (B.G.); rickyjavierrh@gmail.com (R.R.); evygaray@gmail.com (E.G.); 2Instituto Colombiano de Medicina Tropical-Universidad CES, Medellín 050001, Colombia; 3Secretaria de Salud del Vaupés, Mitú 470001, Colombia; hollmanmiller@gmail.com; 4Organización SINERGIAS Alianzas para la Vida, Bogotá 110111, Colombia; mc.rodriguez.vdh@gmail.com; 5Fundación GAIA Amazonas, Bogotá 110111, Colombia; whilkend@gmail.com; 6Corporación Colombiana de Investigación Agropecuaria-Agrosavia, Centro de Investigación Turipaná, Cereté 230550, Córdoba, Colombia; moviedo@agrosavia.co

**Keywords:** public health, population groups, informal sector, asymptomatic diseases, seroepidemiologic studies

## Abstract

**Introduction.** COVID-19 is a pathology caused by the SARS-CoV-2 virus. The World Health Organization (WHO) has reported more than 225 million cases and 4.5 million deaths worldwide. **Objective:** To describe the seropositivity, spatial distribution, and clinical and sociodemographic variables of SARS-CoV-2 in a community of the Colombian Amazon region. **Methods**. In December 2020, a cross-sectional observational study was carried out in a population located in the Colombian Amazon in the municipality of Mitú. Sociodemographic and clinical data were taken. Besides, 589 blood samples were taken, and an antibody detection was carried out with an ELISA and a recombinant protein N antigen of SARS-CoV-2. **Results**. A seropositivity of 57.6% was observed. The highest proportion of the infection is located in inter-municipal transport zones. The bivariate analysis did not show differences in the SARS-CoV-2 infection rate concerning the variables sex, age range, and the presence of comorbidities (*p* > 0.05). The bivariate and multivariate analysis showed that being symptomatic and presenting neurological manifestations of the upper respiratory tract are clinical variables associated with SARS-CoV-2 infection (*p* < 0.05). One of the causes of this virus’s high spread in this community could be that 53.3% of the people were asymptomatic. **Conclusions.** Our data showed a high burden and transmission of SARS-CoV-2 in the indigenous community. This could be linked to cultural behaviors and the high infection rate in asymptomatic patients.

## 1. Introduction

The World Health Organization (WHO) has reported more than 225 million cases and 4.5 million deaths related to COVID-19 in the world [1]. At present, in a short time, the clinical characteristics and environmental factors related to the spread of SARS-CoV-2 in the general population have been studied in depth [2]. However, little is known about this infection in populations of indigenous descent.

Colombia is one of the largest populations with indigenous heritage, and some of them are found in the Amazon. More than 64 indigenous ethnic groups have been described, and 58% of the inhabitants of this region belong to or have a genetic ancestry related to one of these ancestral communities [3,4].

On the other hand, these populations have certain cultural conceptions against this new virus, linked to the low use of protection measures. Therefore, there would be a high proportion of asymptomatic individuals that would facilitate the spread of SARS-CoV-2. However, it is unknown in Colombia and neighboring countries, such as Brazil, Peru, and Venezuela, the behavior and spread of this virus in indigenous communities.

This work’s objective was to describe the seropositivity, spatial distribution, and clinical and sociodemographic variables related to SARS-CoV-2 infection in a population of indigenous ancestry from the municipality of Mitú, Vaupés, Colombian Amazon.

## 2. Methods

Type of study, geographic location, and size of the sample. In December 2020, a cross-sectional observational study was carried out. The study was carried out in the municipality of Mitú, department of Vaupés, a region that occupies part of the great Colombian Amazon [5]. The department is located in the South East of Colombia on the border with Brazil (Figure 1). Five hundred and ninety people were included, the size of the sample was calculated based on the Mitú population of 16,580, with a reliability of 95% and a margin of error of 4% (EPIDAT 3.1).

Ethical aspects. The research was carried out following the international ethical standards given by the WHO, the Pan American Health Organization, and supported by the Declaration of Helsinki. The project also followed the national legislation, resolution number 008430 of 1993 of the Ministry of Health of Colombia, that regulates the studies in health. Furthermore, this work was endorsed by the ethics committee of the Tropic Biological Research Institute.

Serology. A blood sample was taken from each person to detect antibodies against SARS-CoV-2. A commercial ELISA (Eurofins-Ingenasa. Madrid) was used, which uses a recombinant N protein for SARS-CoV-2. The test measures totaled IgG, IgM, and IgA antibodies and were previously validated in our laboratory [6]. Analysis of data. Each individual was asked sociodemographic and clinical aspects; only one person was chosen for each household, and it was not considered whether they had suffered from the SARS-CoV-2 disease. For the sociodemographic and clinical characteristics registry, an epidemiological survey was used by the Biological Research Institute of the Tropics. Data were analyzed using the Statistical Package for the Social Sciences version 27 (SPSS). The univariate analysis for the qualitative variables was carried out through the calculation of absolute and relative frequencies. The measures of central tendency were calculated as quantitative variables.

Further, the normality of the quantitative variables was determined by applying the Kolmogórov–Smirnov test. The bivariate analysis of the qualitative variables was carried out through Pearson’s chi-square test. The qualitative and quantitative variables analysis was done through the Mann–Whitney or Kruskal–Wallis U test if the qualitative variable was polytomous. Multivariate analysis was performed through binomial logistic analysis. The significance of the *p*-value was established at <0.05. The risk analysis was made by calculating the Odds Ratio (OR), and a confidence interval (CI) was included. In addition, a map of points was made for the geolocation of the serological data using the QGIS software of the geographic information system version 3.4.15.

## 3. Results

Sociodemographic and clinical characteristics. Of 589 people, a seropositivity of 57.6% was obtained (Table 1). A total of 56.9% were female gender, and 43.2% were male. A total of 48.1% were in the age range between 20–39 years, and 16.8% had comorbidities, mainly arterial hypertension (27.3%) and diabetes mellitus (17.2%). A total of 27.3% of people presented in the last three months pulmonary or extrapulmonary clinical manifestations associated with COVID-19. Of these people with previous symptoms, it was confirmed that 21.7% had one or more sick relatives with SARS-CoV-2 (Table 1).

Spatial distribution of SARS-CoV2 in the municipality of Mitú. Analysis of the point map showed a distribution of cases throughout the municipality. However, the cases were mainly concentrated in areas close to the airport terminal (Figure 1).

Seropositivity for SARS-CoV2 and gender. No statistically significant differences were found between the percentage of infection between men and women (*p* > 0.05) (Table 2, Figure 2). However, a high proportion of infection was evidenced in both genders (>55%).

Relationship between seropositivity and age ranges. There were no differences between the percentage of SARS-CoV-2 infection in the different age groups (*p* > 0.05) (Table 2, Figure 3). Except for the group ≥ 70 years, all the population groups presented seropositivity > 50%.

Seropositivity and comorbidities. No statistically significant differences were found with seropositivity between individuals with some comorbidity and those who had none. However, 50% of the individuals with comorbidities had been exposed to this new coronavirus (Table 3, Figure 4).

Seropositivity to SARS-CoV-2 and history of clinical manifestations related to COVID-19. Higher seropositivity and elevated antibody titers against SARS-CoV-2 were observed among individuals who had a history of pulmonary and extrapulmonary clinical manifestations in the last three months (*p* < 0.05) (Table 3, Figure 5). A total of 53.3% of the asymptomatic individuals were seropositive.

### Variables That Explain the Seropositivity for SARS-CoV-2 in This Population

The multivariate analysis through binomial logistic regression and the history of symptoms of this disease is the variable that best explains that an individual is seropositive against the virus (*p* < 0.05). These individuals’ risk for being seropositive is almost twice that of the rest of the general population (OR 1.9. 95% CI 1.3–2.9) (Table 4). Considering that being symptomatic was an explanatory variable, it was decided to determine which clinical manifestations best-explained seropositivity. The multivariate analysis showed that the upper respiratory tract’s neurological disorders, such as anosmia and ageusia, are clinical manifestations of this infection’s risk factor (*p* < 0.05) (OR 2.3. 95% CI 1.1–4.7) (Table 5).

## 4. Discussion

The seropositivity against SARS-CoV-2 evidenced in this area of the Colombian Amazon was 57.6% and could be considered within the serological studies as one of the world’s highest. Our findings exceed those found in the principal city of the Brazilian Amazon, Manaus, where 44% community seropositivity was reported [7], and in Atahualpa Ecuador [8], with 45%. At the local level, this infection rate is comparable to that observed in the city of Montería, where we show high community immunity against this new virus (55%) [9].

The reasons that could explain this high seropositivity against SARS-CoV-2 in the municipality of Mitú could be related to the typical cultural behaviors in these populations, such as sharing household utensils, having numerous family nuclei, and engaging in agricultural economic activities [10]. Besides, Mitú is considered a small city of 16,580 inhabitants, and all these factors could influence the free circulation of SARS-CoV-2 and the high infection rate observed in this population. However, it is impossible to differentiate if these SARS-CoV-2 infections are active or an immunological trace of this infection since we evaluate total antibodies and not the active search for cases.

Regarding the spatial distribution of this infection, it was observed that the formal and informal trade zones and residential areas adjacent to inter-municipal transport zones were hotspots for SARS-CoV-2 infection in this population. These findings agree with Liu et al. [11], in which it was observed that in these areas, the spread of SARS-CoV-2 is favored, mainly if biosafety measures, such as social distancing, use of masks, and handwashing, are not applied. 

Regarding the sex of the population evaluated, no differences were found in the SARS-CoV-2 infection rate. However, the high proportion of infected individuals in both genders is striking, higher than 55%, and reflects an essential spread of this virus in men and women in Mitú.

On the other hand, this finding makes us reconsider one of the paradigms of this disease: the greater susceptibility of SARS-CoV-2 infection in males. Testosterone has been reported to promote androgen receptors at the nuclear level that facilitate the expression of TMPRSS2 receptors on the cell membrane, thereby promoting the entry of SARS-CoV-2 into the host cell [12,13]. The male hormone can induce a greater expression of ACE2 receptors at the cardiac and renal levels, another receptor involved in this infection’s pathogenesis [12]. Therefore, it would be necessary to analyze this topic more carefully. In the indigenous and other populations, we did not find evidence that the SARS-CoV-2 infection rate is affected by the sex variable [6].

Regarding the evaluated age ranges, we did not find differences between the percentage of infection with SARS-CoV-2. This finding is consistent with what was found in Sergipe in Brazil, where no differences were found in the seropositivity rate concerning age ranges [14]. However, the infection rate in these Brazilian age groups does not exceed 15%. While in the present study, the population of indigenous ancestry has an infection rate higher than 48%, it could be related to a higher infection rate in the ages 20–39 years (59.5%) because they are the predominantly productive age group. A total of 60% of this population is engaged in informal economic activities [15], in which it is complicated to maintain self-care and isolation measures. Therefore, these infected individuals may have spread the virus to individuals of their family nuclei belonging to other age groups.

When comparing whether individuals with comorbidities presented greater seropositivity against SARS-CoV-2, it was not found that having hypertension or diabetes was related to an increase in the infection rate. However, it is essential to mention that suffering from comorbidities is related to greater severity and prevalence of complications, such as acute respiratory distress syndrome, acute kidney injury, and septic shock [16].

The multivariate analysis showed that the presence of pulmonary and extrapulmonary symptoms of COVID-19 is the variable that best explains seropositivity against SARS-CoV-2 in an indigenous population. 

Furthermore, it was found that clinical manifestations of the upper respiratory tract, such as anosmia and ageusia, are clinical variables that can be pathognomonic of SARS-CoV-2 infection in this population, and these findings have also been reported in the general population [16]. However, it is essential to note that 53.3% of asymptomatic individuals had antibodies against SARS-CoV-2. This could be one of the principal explanations for the high seropositivity and widespread SARS-CoV-2 in this population. However, the present study’s serological and clinical findings would be circumscribed mainly by the dynamics of the SARS-CoV-2 infection in the Mitú population. In other words, it is complex to extrapolate the data of this work with other indigenous communities of the Amazon, taking into account the ethnic plurality and genetics of these populations. However, despite this ethnic plurality and limitations in terms of mobility in the Amazon rainforest, the WHO has determined that the SARS-CoV-2 pandemic has disproportionately affected these indigenous peoples and has estimated that these populations are 40% more vulnerable than the non-indigenous population [17].

In conclusion, our data showed a high burden and transmission of SARS-CoV-2 in the indigenous Mitú community in 2020. Therefore, these findings show that it is necessary to strengthen sanitary services for future outbreaks related to this coronavirus or other viruses. Finally, it is essential to continue studying how this infection and other diseases compromise and impact the public health of these ancestral communities. 

## Figures and Tables

**Figure 1 tropicalmed-06-00191-f001:**
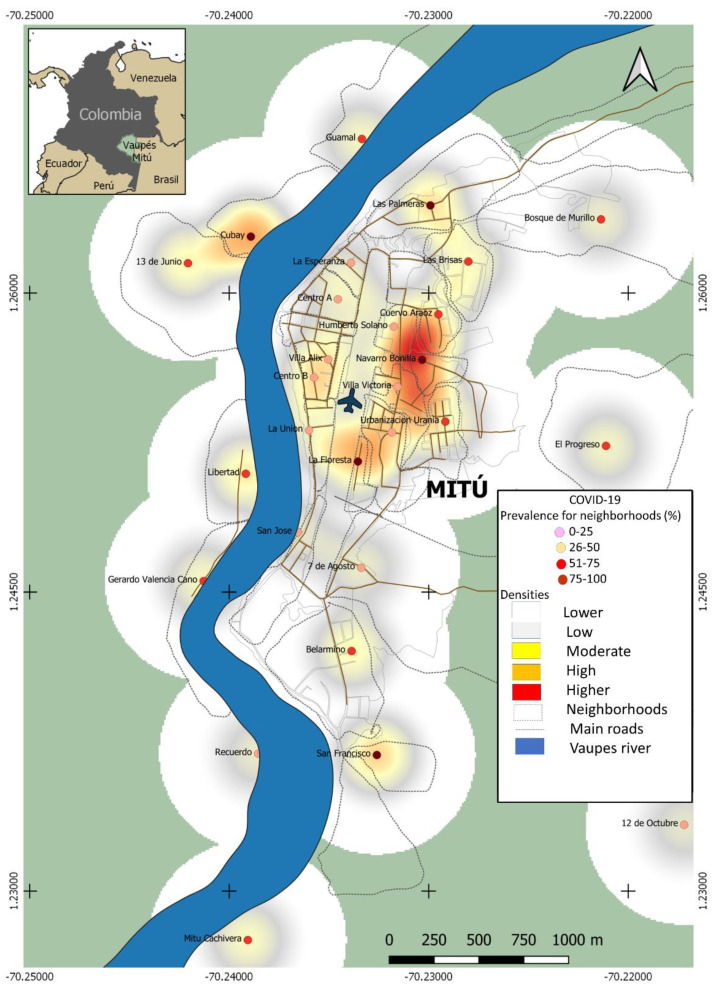
Spatial distribution through point map for SARS-CoV-2 infection in this population. Distribution of SARS-CoV2 cases in Mitú; relevant areas of concentration of infected by SARS-CoV2 close to the airport terminal is observed.

**Figure 2 tropicalmed-06-00191-f002:**
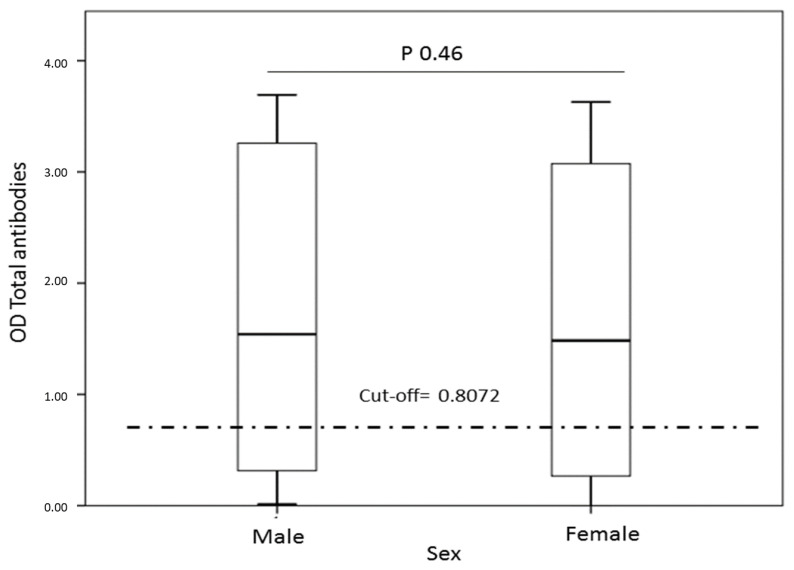
Relationship between sex versus serological data for SARS-CoV-2. A box made up of the 25th, 50th percentiles (median) is observed; 75th and extreme values represent outliers. There is no difference in seropositivity or OD of total antibodies against SARS-CoV-2 between men and women. More than 50% of the data is well above the cut-off.

**Figure 3 tropicalmed-06-00191-f003:**
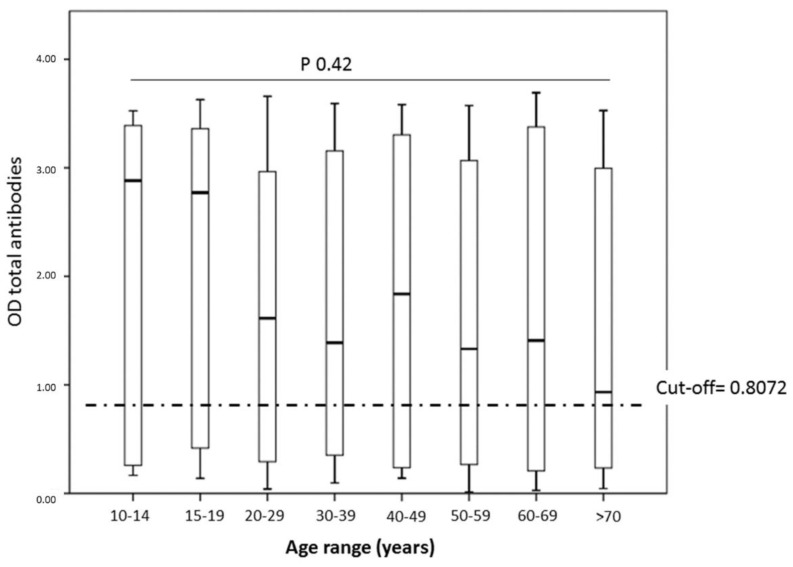
Relationship between the age ranges evaluated against the serological data for SARS-CoV-2. No statistically significant differences were observed when comparing the kinetics of total antibodies against the different age groups.

**Figure 4 tropicalmed-06-00191-f004:**
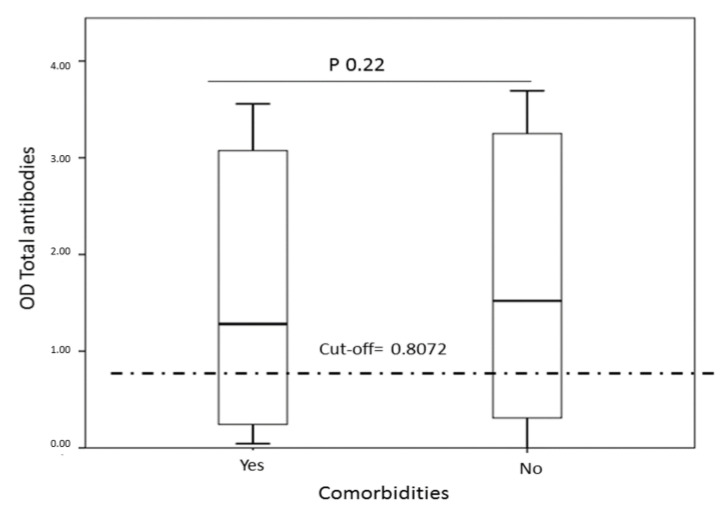
Relationship between having some comorbidity versus the serological data for SARS-CoV-2. No statistically significant differences were observed when comparing the total antibody kinetics between individuals with comorbidity versus those who did not.

**Figure 5 tropicalmed-06-00191-f005:**
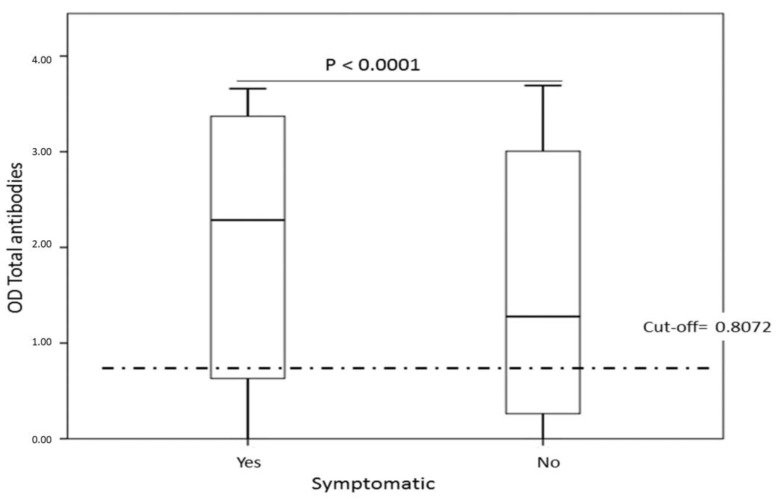
Relationship between the presence of clinical manifestations and a positive serological test for SARS-CoV-2. Higher seropositivity and OD of total antibodies are observed in individuals with clinical manifestations related to SARS-CoV-2 infection.

**Table 1 tropicalmed-06-00191-t001:** Description of the sociodemographic and clinical characteristics of the study participants.

**Characteristic of the Individuals**	N = 589 (%)
**Sex**	
Female	335 (56.9)
Male	254 (43.2)
**Age range (years)**	
10–14	7 (1.2)
15–19	50 (8.5)
20–29	153 (25.9)
30–39	131 (22.2)
40–49	63 (10.7)
50–59	69 (11.7)
60–69	61 (10.3)
≥70	31 (5.3)
**Comorbidities**	
Yes	99 (16.8)
**Comorbidities related**	
Arterial hypertension	27 (27.3)
Diabetes mellitus (DM)	17 (17.2)
Arterial hypertension + DM	6 (6.1)
Others	49 (49.4)
**Some family members diagnostic with COVID-19**	
Yes	128 (21.7)
**Symptomatology related to COVID-19**	
Yes	161 (27.3)
**Symptoms associated**	
Fever	120 (20.4)
Anosmia-ageusia	84 (14.3)
Cough	82 (13.9)
Seropositivity in Mitú (95% CI)	339 (57.6%) (54.2–60.8)

**Table 2 tropicalmed-06-00191-t002:** Relationship between seropositivity against SARS-CoV-2 versus sex and age ranges in years.

Variable	Serology against SARS-CoV-2		*p*-Value
Sex	Positive (%)	Negative (%)	
Female	196 (59)	137 (41)	0.74
Male	142 (55.9)	112 (44.1)	
No data	1 (50)	1(50)	
**Age Range (years)**	**Positive (%)**	**Negative (%)**	***p*-Value**
0–4	0	0	0.58
5–9	0	0	
10–14	4 (57.1)	3 (42.9)	
15–19	34 (68)	16 (32)	
20–29	91 (59.5)	62 (40.5)	
30–39	78 (59.5)	53 (40.5)	
40–49	34 (54)	29 (46)	
50–59	35 (50.7)	34 (49.3)	
60–69	32 (52.5)	29 (47.5)	
≥70	15 (48.4)	16 (51.6)	
No data	16 (66.7)	8 (33.3)	

**Table 3 tropicalmed-06-00191-t003:** Seropositivity against SARS-CoV-2 with comorbidities and symptoms of COVID-19.

Variables	Serology against SARS-CoV-2		*p*-Value
Comorbidities	Positive (%)	Negative (%)	
Yes	51 (51.5)	48 (48.5)	0.39
No	287 (58.8)	201 (41.2)	
No data	2 (66.7)	1 (33.3)	
**Symptoms Related to COVID-19**	**Positive (%)**	**Negative (%)**	***p*-Value**
Yes	110 (68.3)	51 (31.7)	0.005
No	219 (53.3)	192 (46.7)	
No data	11 (61.1)	7 (38.9)	

**Table 4 tropicalmed-06-00191-t004:** Multivariate analysis among the cluster of variables evaluated with the probability of being seropositive for SARS-CoV-2.

Multivariate Analysis	Seropositivity	
	*p*-Value	OR 95% CI
Neighborhood	0.8	0.99 (0.97–1.02)
Sex	0.45	1.1 (0.8–1.6)
Age range	0.15	1.1 (0.97–1.2)
Symptoms	0.002	1.9 (1.3–2.9)
Comorbidities	0.27	0.75 (0.5–1.2)

**Table 5 tropicalmed-06-00191-t005:** Multivariate analysis among the cluster of clinical manifestations associated with COVID19 with the probability of being seropositive for SARS-CoV-2.

Multivariate Analysis	Seropositivity	
	*p*-Value	OR 95% CI
Anosmia/ageusia	0.028	2.3 (1.1–4.7)
Fever	0.75	0.96 (0.8–1.2)
Dyspnea	0.823	0.9 (0.4–1.9)
Headache	0.1	1.7 (0.88–3.3)
Shortness breath	0.68	0.84 (0.4–1.6)
Cough	0.96	0.98 (0.48–1.9)
Diarrhea	0.56	1.2 (0.58–2.7)
Nauseas/vomiting	0.2	0.6 (0.3–1.3)
Myalgia	0.7	0.9 (0.4–1.8)

## Data Availability

The raw data supporting the conclusions of this article will be made available by the authors, without undue reservation, to any qualified researcher. https://docs.google.com/spreadsheets/d/1WNUGbWXByfmjpYjlVya8GoWlXo3SLPwf/edit?usp=sharing&ouid=106067607436651270077&rtpof=true&sd=true (22 October 2021).

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
