# Peer review of "High Prevalence of SARS-CoV-2 in an Indigenous Community of the Colombian Amazon Region"

_tropicalmed, 2021, doi:10.3390/tropicalmed6040191_

Round 1
Reviewer 1 Report
The authors report an amazing finding, which correlates with other studies of tribal people of the Amazon, but not with other studies, like Greek University staff, Cambodian people, and more, all ranging between 1 and 10 percent positive ELISA tests. This makes one think that the sensitivity and the specificity of the test have not been properly validated. The authors refer to reference #6, but this gets you to the company's site, where prices are mentioned, not qualities or validation of the test. The lack of significant differences found in the demographic data further weakens the findings.
Another issue is the mobility of the population, its outside contact, one wants to know a lot more about why this population differs from others. Testing other populations in Colombia with the same ELISA would be interesting, and support the results.
Author Response
Reviewer#1.
- “The authors report an amazing finding, which correlates with other studies of tribal people of the Amazon, but not with other studies, like Greek University staff, Cambodian people, and more, all ranging between 1 and 10 percent positive ELISA tests. This makes one think that the sensitivity and the specificity of the test have not been properly validated. The authors refer to reference #6, but this gets you to the company's site, where prices are mentioned, not qualities or validation of the test. The lack of significant differences found in the demographic data further weakens the findings”.
Reply. According to the comment, we decided to change reference No. 6 to endorses and demonstrate that our research group has already validated the ELISA test used in this work. Please see reference 6.
- “Another issue is the mobility of the population, its outside contact, one wants to know a lot more about why this population differs from others.
Reply. According to the comment, we decided to mention why the mobility aspect and how these populations of indigenous ancestry differ from others. Please see the discussion, lines 224-228.
“Testing other populations in Colombia with the same ELISA would be interesting, and support the results”.
Reply. Our research group has already tested other Colombian populations using this same ELISA kit. Therefore, our findings would be supported by this research "Garay E, Serrano-Coll H, Rivero R, Gastelbondo B, Faccini-Martínez Á, Berrocal J, et al. SARS-CoV-2 in eight municipalities of the Colombian tropics: high immunity, clinical and sociodemographic outcomes. Trans R Soc Trop Med Hyg. 2021 June 29; trab094 ".

Reviewer 2 Report
See attached pdf.

Author Response
- “The authors state that they want to describe the seroprevalence of this population, but instead they are describing the seropositivity of their sample. The difference being that the ‘seropositivity of the sample’is the proportion of people who tested positive, while the underlying ‘seroprevalence of the population’ should be an estimate that accounts for the sampling design, the sensitivity and specificity of the test, and the demographics of the population and includes a measure of uncertainty (i.e., confidence intervals).
While the overall results likely would not change much due to the high seropositivity across subgroups, I do believe it is important to set a precedent of conducting proper seroprevalence analyses”.
Reply. Taking into account the suggestion, we decided on the objective of the manuscript to eliminate the term seroprevalence and replace it with seropositivity and add the 95% confidence interval as a measure of uncertainty. Please see the introduction, line 48, table 1 (Seropositivity in Mitú [95% CI])
- There are too many subgroups in the subanalyses.
“In the age analysis, the 5-9 subgroup has only 1 member. It would be better to have larger samples. For example, members under 20 could be pooled as a school-aged group, 20-50 as working-age, and 50+ as elderly. These should be chosen to match the life-style qualities of the population”.
Reply. We decided to work with these age ranges since the World Health Organization guidelines for serological studies recommend this type of study this stratification " World Health Organization. (2020). Population-based age-stratified seroepidemiological investigation protocol for coronavirus 2019 (COVID-19) infection, 26 May 2020, version 2.0. World Health Organization. https://apps.who.int/iris/handle/10665/332188. Licencia: CC BY-NC-SA 3.0 IGO”.
“The comorbidites group is possibly large enough to be a subgroup (n=99), but the largest individual comorbidity has only 27 members. That could be kept in the descriptive analysis, but is likely too small to find any statistical association”.
Reply. Considering the comment and that the main comorbidity has such a small number of individuals (n = 27), we prefer to keep the analysis carried out for this variable.
“A breakdown of the individual symptoms would be interesting in the descriptive table, but less so in the statistical analyses due to small samples”.
Reply. According to the suggestion, we break down the main clinical manifestations reported by the participants of this study. Please see table 1.
- “I think that symptoms should be removed from the multivariate analysis. The two main reasons for conducting the multivariate analysis could be (1) to extrapolate to the general population or (2) to try and figure out which communities were infected most after holding other factors constant. Without the symptom incidence in the general population, extrapolating those results to the larger population would be impossible (this might also be true for comorbidities). Since symptoms occur after infection, we would not want to account for them in the multivariate analysis”.
Reply. Thanks for the comment. However, we consider evaluating these clinical variables (symptoms and comorbidities) in the multivariate analysis. Given that, we as researchers were not apparent at the beginning of this investigation if the symptoms would be an explanatory variable for the seropositivity found in this study, taking into account the tremendous impact that asymptomatic patients have on this infection.
“The map seems overly ambitious and likely inaccurate. Much of the white and grey areas are probably not low density (I took this to mean low seroprevalence, but it was not clear), but rather unknown seroprevalence. In some places, the color seems like it only represents that it was sampled a lot, rather than actually having a high seroprevalence. It’s hard to tell from the color scheme if the neighborhoods to the east of the airport have seroprevalences between 0-25 or 26-50, but either way, I would expect those to be lower than some of the grey/white areas. It’s interesting to see where the neighborhoods are located, but I think the interpolation is problematic”.
Reply. We appreciate the comment. However, we consider that the map is not inaccurate, taking into account that the samplings throughout the localities of this population were carried out based on the population density in each of them. In addition, the areas that show a lower density of cases correspond to dispersed rural areas adjacent to the evaluated urban area.
- “Please share your code and data. If the data cannot be shared, please provide simulated data”.
Reply. In the following link, the raw data of the investigation is shared. https://docs.google.com/spreadsheets/d/1WNUGbWXByfmjpYjlVya8GoWlXo3SLPwf/edit?usp=sharing&ouid=106067607436651270077&rtpof=true&sd=true

Round 2
Reviewer 1 Report
The article in its present form can be published.
Author Response
The reviewer did not comments.

Reviewer 2 Report
See attached file

Author Response
Reviewer(s)' Comments to Authors:
Reviewer#2.
- “This looks good, though I still see the term ‘seroprevalence’ on lines: 17, 22, 89, 127, 137, 163, 164, and 166”.
Reply. According to your suggestion, we changed the term “seroprevalence” to “seropositivity”. Please see lines, 17, 22, 88, 126, 136, 162, 163, 165.
- “According to the document at the above link, “crude age-specific estimates will need to be adjusted for age structures in the population”. While the document says that it would be ideal to have those age ranges, the data is insufficient to support it, especially at the younger age levels and possibly at the older age levels too.
As noted in other sections before, I don’t think this will tangibly affect the overall conclusions since all of the seropositivity rates are in the same range. It’s just that running a statistical analysis where one group has a single occupant looks and feels wrong to this statistician”.
Reply. According to your suggestion, we decided to eliminate from the analysis the individual evaluated in the age range of 5-9 years and in this way, we better support our statistical analysis. Please see figure 3, table 2.
- “Perhaps the authors could include this is the data availability statement and allow read access for people with the link”. Reply. According to your suggestion, we added the link with raw data of this research. Please see lines 251, 252.

This manuscript is a resubmission of an earlier submission. The following is a list of the peer review reports and author responses from that submission.